# Fine-tuning foundational models to code diagnoses from veterinary health records

**Mayla R. Boguslav[1,2]☉, Adam Kiehl[3]☉\*, David Kott[1]☉, George Joseph Strecker[3]‡, Tracy L. Webb[3], Nadia Saklou[3], Terri Ward[4], Michael Kirby[1]‡**

**1** Data Science Research Institute, Colorado State University, Fort Collins, Colorado, United States of America, **2** Southern California Clinical and Translational Science Institute, University of Southern California, Los Angeles, California, United States of America, **3** College of Veterinary Medicine and Biomedical Sciences, Colorado State University, Fort Collins, Colorado, United States of America, **4** Veterinary Teaching Hospital, Colorado State University, Fort Collins, Colorado, United States of America

☉ These authors contributed equally to this work.
‡ These authors also contributed equally to this work.
\* adam.kiehl@colostate.edu

## Abstract

Veterinary medical records represent a large data resource for application to veterinary and One Health clinical research efforts. Use of the data is limited by interoperability challenges including inconsistent data formats and data siloing. Clinical coding using standardized medical terminologies enhances the quality of medical records and facilitates their interoperability with veterinary and human health records from other sites. Previous studies, such as DeepTag and VetTag, evaluated the application of Natural Language Processing (NLP) to automate veterinary diagnosis coding, employing long short-term memory (LSTM) and transformer models to infer a subset of Systemized Nomenclature of Medicine - Clinical Terms (SNOMED-CT) diagnosis codes from free-text clinical notes. This study expands on these efforts by incorporating all 7,739 distinct SNOMED-CT diagnosis codes recognized by the Colorado State University (CSU) Veterinary Teaching Hospital (VTH) and by leveraging the increasing availability of pre-trained language models (LMs). 13 freely available pre-trained LMs (GatorTron, MedicalAI ClinicalBERT, medAlpaca, VetBERT, PetBERT, BERT, BERT Large, RoBERTa, GPT-2, GPT-2 XL, DeBERTa V3, ModernBERT, and Clinical ModernBERT) were fine-tuned on the free-text notes from 246,473 manually-coded veterinary patient visits included in the CSU VTH's electronic health records (EHRs), which resulted in superior performance relative to previous efforts. The most accurate results were obtained when expansive labeled data were used to fine-tune relatively large clinical LMs, but the study also showed that comparable results can be obtained using more limited resources and non-clinical LMs. The results of this study contribute to the improvement of the quality of veterinary EHRs by investigating accessible methods for automated coding and support both animal and human health

**Data availability statement:** The data used to fine-tune foundational models for this study are proprietary medical records that are the property of Colorado State University; they cannot be shared publicly due to restrictions relating to safeguarding potentially identifying or sensitive patient information. Please contact the Office of the Vice President for Research at vpr_office@mail.colostate.edu for questions or requests regarding access to study data. The fine-tuned GatorTron model is freely available, subject to a data usage agreement, license, and a short CITI training, at https://physionet.org/content/vet-diagnosis-coding/1.0.0/. All code used for data pre-processing and model training is freely available at https://github.com/adam-kiehl/DiagnosisCoding.

**Funding:** This work was partially supported by the National Institute of Health National Center for Advancing Translational Sciences (NCATS; https://ncats.nih.gov/) (salary support to NS through U01TR002953-05) and by the Colorado Clinical and Translational Sciences Institute (CCTSI; https://cctsi.cuanschutz.edu/) (salary support to AK, GS, and TLW through UM1 TR004399). Publication fees were partially supported by the Koster Endowment. Contents are the authors' sole responsibility and do not necessarily represent official NIH views.

**Competing interests:** The authors have declared that no competing interests exist.

research by paving the way for more integrated and comprehensive health databases that span species and institutions.

## Author summary

In this study, we explored the use of advanced natural language processing (NLP) techniques to improve the quality and interoperability of veterinary medical records. By leveraging a variety of pre-trained language models (LMs) and a labeled training dataset curated by expert medical coders to apply standardized medical terminologies to diagnoses from free-text clinical notes, we demonstrate a powerful use-case for recent developments in NLP technologies. Our findings suggest that complex LMs fine-tuned on large volumes of curated data yield best results for quick and reliable automated diagnosis coding. However, we also show that comparable results can be attained using a more minimal set of computational and data resources. We believe this study can provide guidance for other clinical sites interested in enhancing the quality of electronic health records in both the veterinary and human domains. Accurate, automated medical record coding methods may facilitate and encourage clinical research and data sharing in the veterinary, human, and One Health contexts.

## Introduction

The use of veterinary medical records for clinical research efforts is often limited by interoperability challenges such as inconsistent data formats and clinical definitions, and data quality issues [1–4]. Clinical coding is used to transform medical records, often in free text written by clinicians, into structured codes in a classification system like the Systemized Nomenclature of Medicine - Clinical Terms (SNOMED-CT) [5,6]. SNOMED-CT is a "comprehensive clinical terminology that provides clinical content and expressivity for clinical documentation and reporting". It is designated as a United States standard for electronic health information exchange and includes clinical findings, procedures, and observable entities for both human and non-human medicine [6]. SNOMED-CT constitutes a hierarchy of standardized codes beginning with a top-level code such as "Clinical Finding" (SNOMED: 404684003) and terminating with a leaf code such as "Poisoning Due to Rattlesnake Venom" (SNOMED: 217659000). This work aims to improve methods for automating the clinical coding of veterinary medical records using hand-coded records from the Veterinary Teaching Hospital (VTH) at Colorado State University (CSU) as a training set.

The coding of electronic health records (EHRs) into standardized medical terminologies allows for storage in interoperable data models such as the Observational Health Data Sciences and Informatics (OHDSI) Observational Medical Outcomes Partnership (OMOP) Common Data Model (CDM) [7]. Worldwide, more than a billion medical records from over 800 million unique patients are stored in the OMOP CDM format [8], which is designed to promote healthcare informatics and network research. The OMOP CDM provides infrastructure that can facilitate necessary data linkages between EHRs both in veterinary and human medicine. Data linkage in

medical research is a powerful tool that allows researchers to combine data from different sources related to the same patient, creating an enhanced data resource. However, several challenges hinder the full potential of its use, such as identifying accurate common entity identifiers, data inconsistencies or incompleteness, privacy concerns, and data sharing complexities [9].

Despite these challenges, complex linkages like those involved in One Health research are a goal of veterinary and human medical institutions. One Health recognizes the interconnectedness of human, animal, and environmental health domains [10] and the importance of considering all three to optimize health outcomes. Integrating data from different sources – such as human medical records, veterinary medical records, and environmental data – is essential for One Health research [9]. Application of data linkages across human, animal, and environmental data can require multiple types of data linkages; household-level linkages between humans and animals involve (1) linking specific humans (patient identification) with their companion animals (animal identification) and (2) linking the human and animal data for comparisons and analyses (human and veterinary EHRs) in a Health Insurance Portability and Accountability Act (HIPAA)-compliant manner across hospitals. For disease linkage, diagnoses need to be clearly identified across EHRs through standardized coding. This work focuses on identifying diagnoses (7,739 SNOMED-CT diagnosis codes) for animals using SNOMED-CT to pave the way for disease linkage.

## Related work

Most veterinary medical record systems do not currently include clinical coding infrastructure. The free text nature of much of the EHR, especially in veterinary medicine, makes manual clinical coding resource-intensive and difficult to scale. Reliable, supplemental, automated coding is feasible using current Natural Language Processing (NLP) methods including symbolic, knowledge-based approaches and neural network-based approaches [1,11,12]. Symbolic AI makes use of symbols and rules to represent and model the standard practice of clinical coders. Neural networks use training data to "learn" how to match a patient's information to the appropriate set of medical codes [1]. Deep learning methods have been applied to clinical coding since around 2017 with the task formulated as a multi-classification problem, concept extraction problem, or Named Entity Recognition and Linking (NER + L) problem (see [13] for NER in EHRs review). How to best integrate knowledge and deep learning methods has not yet been determined and depends on the purpose of an automated clinical coding system [1]. No matter the purpose, it is important that coders are involved from model development discussions to prospective deployment to ensure quality and usability [2]. Expert clinical coders can provide insight into the large coding systems and complex EHRs to help with automation. SNOMED-CT contains more than 357,000 health care concepts [6], and EHRs contain many different types of documents and subsections, all of varying lengths and formats. How to best represent this varied data in an automated coding context is challenging. Research is ongoing to learn useful dense mathematical representations of patient data that accurately capture meaningful information from EHRs (e.g., [14]).

Much of this ongoing work is focused on creating foundational language models (LMs) for different domains through learning mathematical representations of large amounts of data (see [15,16] for LMs in biomedicine). These LMs can then be fine-tuned for specific downstream tasks including clinical coding. For human medical data, Medical Information Mart for Intensive Care III (MIMIC-III) [17] discharge summaries constitute a prominent publicly available foundational dataset used for benchmarking (MIMIC-IV [18] has recently become available). Several prominent clinical LMs have been trained using this data (e.g., [19–21]). Other research groups have trained clinical LMs on private, proprietary data either in addition to, or in place of, existing foundational data (e.g., [19,22–28]). No freely available foundational datasets like MIMIC-III for the veterinary space were found. Therefore, all current veterinary clinical LMs have been trained on private, proprietary data [29–31]. This may be one contributing factor to the relative dearth of available veterinary clinical LMs.

**Motivating work.** Veterinary-specific automated clinical coding work has been done previously by other groups using a large set of manually coded veterinary medical record data from the CSU VTH [29,32,33]. Both DeepTag [32]

and VetTag [29] used the text from 112,557 veterinary records hand-coded with SNOMED-CT diagnosis codes by expert coders at the CSU VTH. Over one million unlabeled clinical notes from a large private specialty veterinary group (PSVG) were additionally used for pre-training. Both models also made use of the SNOMED-CT hierarchy "to alleviate data imbalances and demonstrate such training schemes substantially benefit rare diagnoses" [29]. The supplied set of target codes for each record were supplemented with each code's hierarchical ancestry to form a hierarchical training objective. This hierarchical prediction approach "first predicts the top level codes and then sequentially predicts on a child diagnosis when its parent diagnosis is predicted to be present" [29]. It was found that utilizing the hierarchy improved performance on both a held-out test set from the CSU VTH labeled data set and a test set from a private practice clinic (PP) in northern California.

DeepTag (2018) is a bidirectional long-short-term memory (BiLSTM) model, which achieved impressive performance (weighted F1 score of 82.4 on CSU and 63.4 on PP data) across a set of 42 chosen SNOMED-CT diagnosis codes. Vet-Tag (2019) expanded on the original concept of DeepTag by employing a transformer [34] model architecture and expanding the size of the set of chosen SNOMED-CT codes to 4,577. Pre-training was performed using the PSVG data and fine-tuning was done on the CSU VTH data. The expanded scope and improved design of VetTag made it the state-of-the-art in the automated veterinary diagnosis coding field (weighted F1 score of 66.2 on CSU VTH and 48.6 on PP data). VetLLM (2023) [33] evaluated a recently developed foundational LM, Alpaca-7B [35] in a zero-shot setting, supplying it 5,000 veterinary notes to identify nine high-level SNOMED-CT disease codes (weighted F1 score of 74.7 on CSU VTH and 63.7 on PP data). The DeepTag, VetTag, and VetLLM trained models and data are not publicly available due to use of proprietary medical data.

**Human clinical LMs.** FasTag [26], a study into the cross-applicability of human and veterinary clinical data, showed that applying human medical data to veterinary clinical tasks has promise considering the limited availability of veterinary LMs. Building on FasTag, four prominent, publicly available human medicine LMs that can be fine-tuned on a variety of downstream tasks including clinical coding were considered and tested for this study: GatorTron (2022) [19], MedicalAI ClinicalBERT (2023) [23], medAlpaca (2023) [28], and Clinical ModernBERT (2025) [36]. Others that were not specifically considered for this study due to computational limitations include MEDITRON (2023) [27] and Me-LLaMA (2024) [21].

GatorTron is one of the largest clinical language models in the world and was created through a partnership between the University of Florida (UF) and NVIDIA. The model was pre-trained on a corpus of >90 billion words of clinically-adjacent text from de-identified UF human clinical notes (2.9 million clinical notes from 2.4 million patients), the publicly-available MIMIC-III dataset, PubMed, and Wikipedia. The model displayed strong improvements, relative to previous models, in five downstream NLP tasks including "clinical concept extraction, medical relation extraction, semantic textual similarity, natural language inference, and medical question answering" [19]. GatorTron is a versatile LM that can be trained for many different tasks, including veterinary clinical coding.

MedicalAI ClinicalBERT is a clinical language model constructed by researchers at Fudan University. It was selected for exploration as a lighter-weight alternative to the massive GatorTron. It should be noted that other models with the name "ClinicalBERT" [20] exist, but to encompass as much data and prior work as possible, a relevant fine-tuned Clinical-BERT model, MedicalAI, was used for this study. The model was pre-trained on 1.2 billion words from clinical records representing a diverse set of diseases. It was then fine-tuned on clinical data from hospitalized patients with Type 2 Diabetes (T2D) who received insulin therapy from January 2013 to April 2021 in the Department of Endocrinology and Metabolism, Zhongshan Hospital and Qingpu Hospital, in Shanghai, China. It was benchmarked on an extraction of 40 symptom labels from clinical notes including free-text descriptions of present illness and physical examination [23], similar to the task of this study.

medAlpaca was inspired by the potential for LMs like those in the GPT series to "[improve] medical workflows, diagnostics, patient care, and education" [28]. It was meant to be a resource that could be locally deployed to protect patient privacy. A dataset named Medical Meadow, aimed at encapsulating a broad set of clinical use-cases, was curated and used to fine-tune the LLaMA [37] foundational model. Medical Meadow consisted of question-answer pairs from Anki

Medical Curriculum flashcards, medically relevant Stack Exchange forums, Wikidoc articles, and various foundational clinical datasets. medAlpaca was benchmarked in a zero-shot setting against portions of the United States Medical Licensing Examination (USMLE) [28].

Clinical ModernBERT [36] is a clinical language model based on the ModernBERT [38] architecture. ModernBERT utilizes *rotary positional embeddings* (RoPE), *unpadding*, and *flash attention* to improve model efficiency. The model was pretrained on over 13 billion tokens from the publicly available MIMIC-IV dataset, PubMed, and ICD-9 through ICD-12 concept descriptions. Clinical ModernBERT outperformed other BERT-based models on EHR, PubMed-NCT, and MedNER classification benchmark tasks.

**Veterinary clinical LMs.** Much prior work has focused on automated clinical coding in human EHRs [1,14,22–26] for insurance claims and other higher level tasks such as enhancing patient outcomes [39]. However, limited clinical coding work has been done in the veterinary space with the lack of a similar financial incentive [29–33]. While not publicly available, DeepTag [32] and VetTag [29] both served as strong inspirations for this study. Two other foundational veterinary clinical LMs that are publicly available and were considered and tested for this study are VetBERT (2020) [30] and PetBERT (2023) [31].

VetBERT aimed to identify the reason or indication for a given drug to aid in antimicrobial stewardship. The VetCompass Australia corpus containing over 15 million clinical records from 180 veterinary clinics in Australia (5% of all vet clinics there) [40] was used for pre-training. A portion of these records were manually mapped to 52 codes in VeNom (Veterinary Nomenclature) [41] for fine-tuning. VetBERT was based on the ClinicalBERT [20] foundational LM and achieved strong predictive accuracy for the downstream task of identification of reason for drug administration.

Most recently, Farrell et al. created PetBERT, a BERT-based LM trained on over five million EHRs from the Small Animal Veterinary Surveillance Network (SAVSNET) in the United Kingdom with 500 million words. The focus was on first-opinion EHRs or primary care clinics that "often lack equivalent diagnostic certainty to referral clinics, frequently drawing upon specialized expertise and a wealth of supplementary diagnostic information" [31]. Further, PetBERT-ICD was created for the specific clinical coding task of classifying EHRs to 20 ICD-11 [42] diagnosis categories, using 8,500 manually annotated records (7,500 training, 1,000 final evaluation). It achieved a weighted average F1 score of 84 in this task [31].

**Non-clinical LMs.** Non-clinical pre-trained LMs are relatively more abundant than those in the clinical domain due to less privacy concerns (e.g., HIPAA). Several prominent and tractably sized non-clinical LMs considered and tested for this study were BERT (2018) [43], RoBERTa (2019) [44], GPT-2 (2019) [45], DeBERTa V3 (2021) [46], and ModernBERT (2024) [38].

Bidirectional Encoder Representations from Transformers (BERT) was introduced by Google in 2018 as a novel encoder-only variation on the original transformer model [34]. It differs from previous transformer models by using masked language modeling (MLM) and next sentence prediction (NSP) pre-training, and by removing the decoder component of the transformer. This procedure makes the model specifically designed for problems involving fine-tuning on a downstream task. Several other models discussed in this paper (MedicalAI ClinicalBERT, VetBERT, and PetBERT) were based on the BERT architecture. BERT was pre-trained on a corpus of 800 million words from BooksCorpus [47] and 2.5 billion words from Wikipedia. It was benchmarked on eleven standard NLP tasks and achieved state-of-the-art results for its time.

It was later determined that the foundational BERT LM was "significantly undertrained;" RoBERTa [44] was offered as a better-constructed alternative. An enhanced pre-training dataset was assembled using the BooksCorpus and Wikipedia datasets used to train BERT as well as CC-News - 63 million scraped news articles from 2016 to 2019, OpenWebText [48] - scraped text from URLs shared on Reddit, and Stories [49] - a subset of the CommonCrawl [50] foundational dataset. With this enhanced dataset and an improved training scheme, RoBERTa achieved state-of-the-art performance on a variety of standard NLP benchmarks [44].

The foundational BERT and RoBERTa LMs were further improved upon by the introduction of Decoding-enhanced BERT with disentangled attention (DeBERTa) [46] and ModernBERT [38]. DeBERTa utilized a *disentangled attention*

*mechanism*, an *enhanced mask decoder*, and a *virtual adversarial training method* to improve model efficiency and performance. It was trained on an English Wikipedia dump, BookCorpus [47], OpenWebText [48], and Stories [49]. ModernBERT utilized *rotary positional embeddings* (RoPE), *unpadding*, and *flash attention* to improve model efficiency. Both model architectures demonstrated superior performance relative to previous BERT-based models.

In contrast to BERT, Generative Pre-trained Transformer 2 (GPT-2) [45] is a decoder-only transformer released by OpenAI. The decoder-only architecture relies on unidirectional pre-training and was designed for a variety of zero-shot tasks. GPT-2 was pre-trained on an OpenAI-generated WebText dataset consisting of over eight million documents scraped from the web in December 2017. Text from Wikipedia was specifically excluded from WebText due to its common usage in other foundational models. It achieved state-of-the-art results on seven standard NLP benchmarks in a zero-shot setting.

### Aims

This study aimed to extend and improve upon prior work in the field of automated clinical coding for veterinary EHRs to facilitate standardization of records, veterinary patient health and research, and the creation of data linkages to support One Health approaches to problem solving. Available manually coded data from the CSU VTH was used to determine if fine-tuning existing foundational models could achieve state-of-the-art results for automated veterinary clinical coding to 7,739 SNOMED-CT diagnosis codes, the largest set of diagnosis codes yet used in a veterinary context. It was found that fine-tuning the foundational model GatorTron performed the best with an average weighted F1 score of 76.9 and an exact match rate of 52.2%.

The primary contributions of this paper are:

- A demonstration that state-of-the-art results can be achieved for clinical coding of diagnoses from veterinary EHRs using the largest number of diagnosis codes to-date through fine-tuning of foundational models (best results from a fine-tuned GatorTron).
- A comparison of current freely available state-of-the-art foundational models for this task.
- A further demonstration of the use of human and general foundational models in the veterinary space (see also FasTag).
- A preliminary performance analysis of the models by frequency of codes, depth of codes in the SNOMED-CT hierarchy, and volume of training data.

## Materials and methods

The automatic coding of data from many clinical domains such as procedures, medications, lab tests, etc., was amenable to straightforward rules-based methods based on concrete mappings between standardized CSU VTH terms and codes from standardized medical terminologies (Fig 1). However, diagnoses were recorded in a free-text fashion (Fig 2) and were not associated with any standard map-able list. This work focused solely on mapping the diagnosis domain to SNOMED-CT.

### Data

The CSU VTH EHRs contain multiple document types including medical summaries, summary addendums, consultations, surgical/endoscopy reports, diagnostic test results, and imaging reports, with each containing a variety of formats and subsections. The medical summary is the main EHR document and contains much of the clinical information for each animal: *presenting complaint*, *history*, *assessment*, *physical exam*, *diagnosis*, *prognosis*, *follow-up plan*, *procedures and treatments*, and others. Information was manually coded directly from the diagnosis lists in the medical summary documents and ancillary reports, if present and applicable. The input for this automated clinical coding task was preprocessed

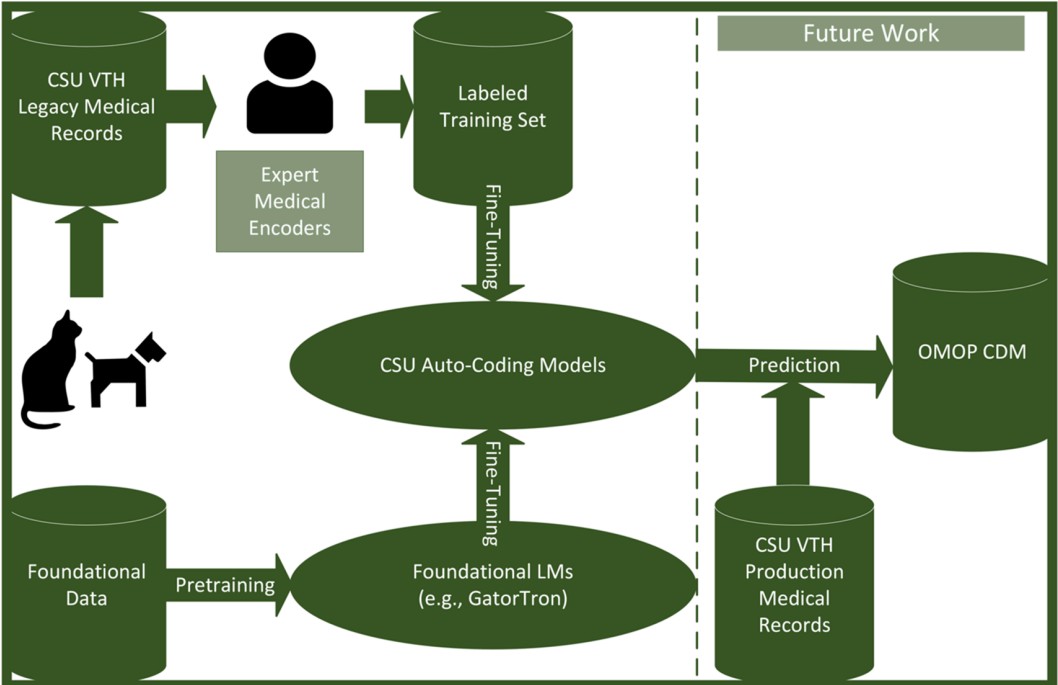

**Fig 1. Colorado State University (CSU) automated clinical coding pipeline.** Raw records are moved to a standardized Observational Medical Outcomes Partnership (OMOP) Common Data Model (CDM) instance, with future work noted. VTH = Veterinary Teaching Hospital; LM = language model.

---

**Clinical Text:**

Diagnoses

-Type I plantar P1 fragments had been diagnosed medially and laterally in the left hind fetlock joint

Assessment

ARTHROSCOPIC SURGERY: The mare was operated on in dorsal recumbency. After surgical preparation and draping the arthroscope was placed through the proximal aspect of the lateral plantar pouch of the left hind fetlock joint. Examination showed chronic proliferated synovial membrane below the lateral sesamoid and this was resected to reveal the fragment, which was embedded in joint capsule. Using the flat sesamoid knife the plantar fragment was elevated and removed with FerrisSmith rongeurs and the defect then debrided with the motorized resector. The arthroscope was then placed to visualize below the medial sesamoid bone. After removal of synovial membrane the fragment was elevated and was quite large. The fragment was severed from the soft tissue attachments with the flat sesamoid knife and removed. It was 1 cm wide and approximately 8 mm thick. After removal of this fragment the defect was debrided and the joint flushed. There was no other damage in the joint.

**Manually Assigned SNOMED-CT Codes:**

81576005 (closed fracture of phalanx of foot)

**Fig 2. Sample excerpt from a CSU VTH medical summary document.**

free text sections from a medical summary for a visit (note that it does not include any other documents). The output for this task was a SNOMED-CT diagnosis list for that visit.

Diagnoses in the CSU veterinary EHRs were manually coded using SNOMED-CT. These efforts were led by the CSU VTH Medical Records team, which was consulted throughout this study for insights into the data and results. 246,473 patient visits featuring 7,739 distinct SNOMED-CT diagnosis codes were manually coded from the legacy EHR system representing visits from 2012 until its replacement in 2019.

SNOMED-CT terminologies are version controlled with older codes being occasionally upgraded or deprecated in favor of new "active" codes. The version of SNOMED-CT used here was downloaded from OHDSI in December 2022. For this study, both active and inactive codes were allowed for practical reasons. The set of diagnosis codes used for manual annotation over many years at the CSU VTH naturally diverged somewhat from being included and active in the latest SNOMED-CT release and includes some (530, 7%) currently inactive codes. There are inactive-to-active mappings available for many codes that can translate outdated vocabulary sets to conform with current standards when needed [51].

**Manual record coding process.** All records were coded by the CSU VTH Medical Records team. The team processed a record once a visit record was closed and all associated documents were finalized (approved by a primary clinician). They began by reviewing the medical summary and invoice for the visit. The medical summary provided information about the diagnoses and the invoice helped confirm what procedures were performed during the visit. The *diagnosis* section of the medical summary was used for coding diagnoses, and other sections, such as *procedures and treatments*, were used to determine the need to review additional documents. For example, surgical reports and invoices were cross-referenced with the medical summary to ensure that procedures were officially performed and should be coded. Laboratory reports (histopathology, necropsy, cultures, fecal samples, etc.) were also checked for additional diagnoses.

**Input fields and preprocessing.** In consultation with the CSU VTH Medical Records team, it was determined that the *diagnosis* section contained the primary information used to inform diagnosis coding. This finding was confirmed by a sensitivity analysis early on (results not shown) that examined model performance for each section and with different combinations of medical summary sections (i.e., *diagnosis*, *assessment*, *presenting complaint*, *history*, *physical exam*, *procedures, and treatments*) used as input text. The only section that independently drove model performance was *diagnosis*. Other sections did not exhibit the capacity for strong independent performance, and their inclusion in the input text alongside *diagnosis* did not significantly improve performance. Only the longer narrative *assessment* section was appended to *diagnosis* for the broad context it provided to visit summaries. Other sections of the EHR were not included due to model restrictions on input text length and in the interest of minimizing text truncation.

Based on the CSU VTH Medical Records team and the early on sensitivity analysis, the final input text for this task was a concatenation of *diagnosis* (including histopathology diagnosis text) and *assessment* sections, in that order. Minimal text cleaning was performed (only simple replacement of Hypertext Markup Language (HTML) characters, lowercase, etc.), and no de-identification methods were used. Records were tokenized according to the custom tokenizers associated with each pre-trained LM and truncated to the corresponding maximum length (truncation rate did not exceed 23% for any tokenizer, records contained 260 tokens on average, and the rate of unknown tokens used did not exceed 0.36% for any tokenizer). Target codes for each record were then one-hot encoded for training. Using a stratified sampling technique to account for class imbalances, the full fine-tuning dataset was split into training (80%), validation (10%), and test (10%) sets.

The output of fine-tuned models was a list of potential diagnosis codes that could be assigned to a record based on the probability of those diagnosis codes being relevant. Note that, unlike VetTag, the SNOMED-CT hierarchy was not considered in outputs but instead was used later for an assessment of code specificity. Raw model outputs for each record were a vector of length 7,739 with logit scores for each class. A Sigmoid activation function was then applied to the raw outputs to yield a vector of probabilities representing the probability that each diagnosis code could be correctly applied to

the record text in question. Predictions were then generated by comparing all probabilities to a chosen prediction threshold (0.5, in accordance with VetTag); a prediction was made when the probability of a code being applied to the record text in question was greater than the chosen threshold.

**Evaluation metrics**

Several key metrics were selected to evaluate the performance of automated coding models. Precision, recall, and F1 were computed through a weighted macro (class-wise) average method to account for class imbalances using the following formulas:

$$P = Number\ of\ diagnosis\ code\ classes$$

$$N = Number\ of\ test\ records$$

$$n_i = Number\ of\ occurrences\ for\ class\ i$$

$$codes_{target,j} = Set\ of\ human-assigned\ codes\ for\ record\ j$$

$$codes_{predicted,j} = Set\ of\ model-predicted\ codes\ for\ record\ j$$

$$Precision_{weighted} = \frac{1}{\sum_{i=1}^{P} n_i} \sum_{i=1}^{P} n_i \times Precision_i \tag{1}$$

$$Recall_{weighted} = \frac{1}{\sum_{i=1}^{P} n_i} \sum_{i=1}^{P} n_i \times Recall_i \tag{2}$$

$$F1_{weighted} = \frac{1}{\sum_{i=1}^{P} n_i} \sum_{i=1}^{P} n_i \times F1_i \tag{3}$$

$$Exact\ Match\ (EM) = \frac{1}{N} \sum_{j=1}^{N} (codes_{target,j} = codes_{predicted,j}) \tag{4}$$

These metrics were chosen in accordance with those reported by the VetTag team and as standard multi-label classification metrics where a skewed class distribution is present [52].

**Models**

Existing state-of-the-art human clinical LMs (GatorTron, MedicalAI ClinicalBERT, medAlpaca, Clinical ModernBERT), veterinary (VetBERT, PetBERT), and general knowledge LMs (BERT, BERT Large, RoBERTa, GPT-2, GPT-2 XL, DeBERTa V3, ModernBERT) were leveraged by fine-tuning them for veterinary clinical coding to 7,739 SNOMED-CT diagnoses (Table 1). Ensemble, mixture of experts, and agentic models were not considered for this study. All models were fine-tuned under a consistent training framework (except for Bert Large, which did not feature a dropout layer) that included a batch size of 32, a binary cross-entropy loss function, an AdamW optimizer, scheduled optimization with 5,000 warmup steps and a plateau learning rate (constant learning rate after warmup) of $3*10^{-5}$, and an early stopping criterion with five epoch patience and a maximum of 50 epochs (number of epochs without improved results required to trigger an early stop). Through experimentation, it was determined that best model performance was achieved when the maximum number of transformer blocks were allowed to update in the fine-tuning process. All transformer blocks for all models were fine-tuned with the exception of GatorTron and medAlpaca, which were too large to fully fine-tune given the computational resource available. No embeddings were fine-tuned. Custom pooler layers consisting of a fully-connected linear layer followed by a Tanh activation function, dropout layers (rate of 0.25), and linear classifier layers were appended to each model. No gradient accumulation or clipping, or progressive unfreezing methods were used. No additional pretraining was

**Table 1**. Selected clinical and non-clinical foundational language models.

| Model | Pre-Training | Parameters | Transformer Blocks | Self-Attention Heads | Embedding Dimension |
|---|---|---|---|---|---|
| GatorTron | Univ. of Florida (2.9M notes), MIMIC-III, PubMed, WikiText (91B total words) | 3.9B | 48 | 40 | 2,560 |
| MedicalAI ClinicalBERT | Zhongshan Hospital, Qingpu Hospital (1.2B total words) | 135M | 6 | 8 | 768 |
| medAlpaca | Anki flashcards, Stack Exchange, Wikidoc, other Q&A | 6.6B | 32 | 32 | 4,096 |
| VetBERT | VetCompass (15M total notes, 1.3B total tokens) | 108M | 12 | 12 | 768 |
| PetBERT | UK Vet EHRs (5.1M total notes, 500M total words) | 108M | 12 | 12 | 768 |
| BERT | BooksCorpus (800M words), Wikipedia (2.5B words) | 108M | 12 | 12 | 768 |
| BERT Large | Same as BERT | 335M | 24 | 16 | 1,024 |
| RoBERTa | Same as BERT with CC-News (63M articles), OpenWebText, Stories | 125M | 12 | 12 | 768 |
| GPT-2 | WebText (8M documents) | 125M | 12 | 12 | 768 |
| GPT-2 XL | Same as GPT-2 | 1.6B | 48 | 25 | 1,600 |
| DeBERTa V3 | Wikipedia, BookCorpus, OpenWebText, Stories | 184M | 12 | 12 | 768 |
| ModernBERT | Web documents, code, scientific literature (2T tokens) | 149M | 22 | 12 | 768 |
| Clinical ModernBERT | MIMIC-IV, PubMed, Clinical Codes (e.g., ICD) (13B tokens) | 136M | 22 | 12 | 768 |
| *VetTag** | *PSVG (1M total notes)* | *42M* | *6* | *8* | *Unknown* |

Models are characterized by pre-training data source and volume, size (number of parameters), number of transformer blocks, number of self-attention heads, and embedding dimension. All models were fine-tuned on Colorado State University Veterinary Teaching Hospital records. M = million; B = billion; UK = United Kingdom; Vet = veterinary; EHRs = electronic health records; BERT = bidirectional encoder representations from transformers; GPT = generative pre-trained transformer; PSVG = private specialty veterinary group.
*Model not fine-tuned for this project and utilized a hierarchical training objective: shown only for comparison.

performed to isolate the effects of fine-tuning on model performance. The results for all fine-tuned models were compared to the results of the current state-of-the-art model in veterinary clinical coding, VetTag. Additionally, all models were tested "out-of-the-box" to determine if it was necessary to fine-tune the models for this task. In this case, all pre-trained weights were frozen, and pooler and classifier layers were fine-tuned under the same conditions described above. Unlike DeepTag and VetTag, a hierarchical training objective [29] was not used in this study.

## Performance analysis

To properly prioritize and caveat findings with the importance of medical diagnosis code assignment, model performance was analyzed to determine the strengths and weaknesses of the model. As initial assessments, performance with respect to code frequency, code specificity (depth in hierarchy), and volume of the fine-tuning data was assessed for the best performing model (i.e., fine-tuned GatorTron). An evaluation of model performance with respect to code frequency informed the amount of data required for each code. Further, an evaluation of model performance with respect to code specificity informed the level of granularity expected of model predictions. To analyze the minimum requirements for the volume of labeled fine-tuning data needed to achieve strong results, model results were collected based on fine-tuning performed using successively smaller fractions of the total training data available.

### Computing resources

All models were fine-tuned on either a DGX2 node or a PowerEdge XE8545 node:

- **DGX2 Node**:
  - 16 Tesla V100 GPUs, each with 32 GB of VRAM
  - One 96-thread Intel Xeon Platinum processor
  - 1.4 terabytes of RAM
- **PowerEdge XE8545 Node**:
  - 4 Tesla A100 GPUs, each with 80 GB of VRAM
  - Two 64-core AMD EPYC processors
  - 512 GB of RAM

### Results

State-of-the-art results were achieved on veterinary clinical coding to the largest set of diagnosis concepts (Table 2). GatorTron fine-tuned for this task achieved the best results with an average weighted F1 score of 76.9 and an exact match rate of 52.2% on a held-out test set (10% of the labeled CSU VTH data). Many smaller models considered for this study (e.g., VetBERT and RoBERTa) still performed comparably to the larger GatorTron (Table 2). Overall, all models had higher precision compared to recall.

The "out-of-the-box" results were poor (Table 3), confirming the need for fine-tuning. On average, models' F1 scores were 52.9 points lower in the "out-of-the-box" setting compared with more expansive fine-tuning. Particularly of note were the BERT and DeBERTa models which failed to converge to a reasonable solution, and the GPT-2 models, which performed strongly relative to all other models. Poorer results in this setting are expected as adapter-based fine-tuning can result in unstable model performance due to a failure to "capture complex data patterns" [53].

**Table 2**. **Results computed on electronic health record text test sets.**

| Model | F1 | Precision | Recall | Exact Match | Fine-Tuning Time |
|---|---|---|---|---|---|
| GatorTron | **76.9**±0.14 | **81.5**±1.74 | **74.4**±1.14 | **52.2**±0.66 | 23.7±0.14 hrs |
| MedicalAI ClinicalBERT | 69.5±1.50 | 79.2±1.15 | 64.4±1.49 | 45.2±1.79 | 1.6±0.25 hrs |
| medAlpaca | 74.0±0.14 | 81.3±0.72 | 70.4±0.50 | 48.3±0.38 | 18.5±2.94 hrs |
| VetBERT | 72.8±0.50 | 79.8±0.50 | 68.9±0.76 | 48.4±0.14 | 3.2±1.23 hrs |
| PetBERT | 71.6±2.58 | 79.4±0.76 | 67.4±4.31 | 47.6±2.35 | 3.2±1.03 hrs |
| BERT Base | 72.2±1.55 | 79.9±0.90 | 67.9±2.23 | 47.7±1.25 | 3.0±0.63 hrs |
| BERT Large | 70.3±5.76 | 78.2±2.41 | 66.0±7.16 | 45.9±4.86 | 8.1±5.22 hrs |
| RoBERTa | 71.5±0.50 | 79.2±1.08 | 67.3±1.27 | 47.0±0.38 | 3.7±1.74 hrs |
| GPT-2 Small | 70.6±1.24 | 79.6±0.29 | 65.8±1.79 | 45.7±1.38 | 4.3±0.38 hrs |
| GPT-2 XL | 73.7±3.31 | 80.5±1.97 | 70.1±3.71 | 48.4±3.37 | 18.5±6.35 hrs |
| DeBERTa V3 | 68.2±3.31 | 77.3±3.58 | 63.7±2.88 | 44.9±1.86 | 7.9±1.17 hrs |
| ModernBERT | 64.1±4.34 | 75.7±1.14 | 58.8±5.65 | 40.4±4.26 | 5.0±0.14 hrs |
| Clinical ModernBERT | 70.8±0.63 | 79.4±0.38 | 66.4±0.86 | 46.8±0.87 | 6.9±0.43 hrs |
| *VetTag** | *66.2* | *72.1* | *63.1* | *26.2* | *Unknown* |

Only *diagnosis* and *assessment* fields from the medical summary were used based on the Colorado State University Veterinary Teaching Hospital Medical Record coders and an early on sensitivity analysis. All manually assigned Systematized Nomenclature of Medicine - Clinical Terms (SNOMED-CT) diagnosis codes for these visits were used regardless of frequency or standard status. Training was not supplemented using the SNOMED-CT hierarchy. Testing was performed three times for each model to allow for the computation of 95% confidence intervals ($\mu \pm t_{.975,2} * \sigma/\sqrt{3}$). Test sets were standardized across models through set random seeds.
*Top result from VetTag publication using 4,577 standard SNOMED-CT diagnosis codes with hierarchically enriched target sets is shown for comparison.

**Table 3**. Out-of-the-box results computed on electronic health record text test sets.

| Model | F1 | Precision | Recall | Exact Match |
|---|---|---|---|---|
| GatorTron | 23.6 | 48.1 | 18.1 | 17.3 |
| MedicalAI ClinicalBERT | 9.7 | 30.5 | 6.9 | 11.2 |
| medAlpaca | 18.7 | 37.7 | 13.7 | 14.8 |
| VetBERT | 12.7 | 33.3 | 9.6 | 13.7 |
| PetBERT | 21.3 | 44.6 | 16.4 | 15.7 |
| BERT Base | 1.4 | 5.6 | 0.9 | 6.7 |
| BERT Large | 0.4 | 6.9 | 0.2 | 5.3 |
| RoBERTa | 5.3 | 17.1 | 4.3 | 10.1 |
| GPT-2 Small | 58.0 | 69.3 | 53.3 | **33.6** |
| GPT-2 XL | **60.7** | **69.6** | **57.1** | 33.4 |
| DeBERTa V3 | 1.1 | 6.3 | 0.6 | 5.2 |
| ModernBERT | 7.4 | 23.4 | 5.9 | 11.9 |
| Clinical ModernBERT | 23.1 | 51.0 | 17.2 | 16.0 |

Foundational model weights were frozen and only pooler and classifier layers were allowed to update during the fine-tuning process. All other training practices were the same as those in Table 1.

## Input formats

A sensitivity analysis with the data input selection for the best model (Table 4) found that while the *diagnosis* and *assessment* sections of a visit record each individually provided some predictive capacity, the *diagnosis* portion was most useful. Note that the raw clinical text was necessarily truncated based on the maximum input length for each model (e.g., there was a 16.7% truncation rate when *diagnosis* and *assessment* were concatenated, in that order, for GatorTron). *Diagnosis* concatenated with *assessment* yielded marginally better results than either section did alone.

## Performance analysis

There was a weakly positive trend between frequency of code usage and predictability ($r = 0.51$; see Fig 3). However, a subset of rarely appearing codes had perfect predictive performance ($n = 398$). Ignoring the extremes, there was little relationship between code frequency and model performance. It should be noted that Tables 4 and 5 and Figures 3-5 are for the fine-tuned GatorTron.

Diagnosis codes were further characterized by their position in the SNOMED-CT hierarchy (Fig 4). Model performance was not observably associated with code specificity ($r_{F1} = r_{Precision} = r_{Recall} = 0.06$). Strong average performance was observed at highly specific depths; however, assessment was limited as these depth categories contained few codes.

Performance was also observed to increase with larger amounts of fine-tuning data ($r_{F1} = 0.62$, $r_{Precision} = 0.50$, $r_{Recall} = 0.66$, $r_{EM} = 0.74$; see Fig 5). All metrics followed a similar pattern with diminishing marginal returns on increased volumes of labeled fine-tuning data: using only 25% of all available labeled fine-tuning data yielded a 10.4% erosion of model performance, according to F1.

**Table 4**. Input field permutation testing.

| Input | F1 | Precision | Recall | Exact Match | Average Tokens | Truncation Rate |
|---|---|---|---|---|---|---|
| Diagnosis | 76.4 | 80.2 | 74.6 | 50.5 | 50 | 0% |
| Assessment | 40.4 | 56.1 | 34.2 | 22.0 | 257 | 12% |
| Diagnosis + Assessment | **76.9** | **81.5** | **74.4** | **52.2** | 260 | 16.7% |

Results from fine-tuning GatorTron on different permutations of the use of *diagnosis* and *assessment* sections of the medical summary. Truncation rate is the percentage of all records that had any truncation.

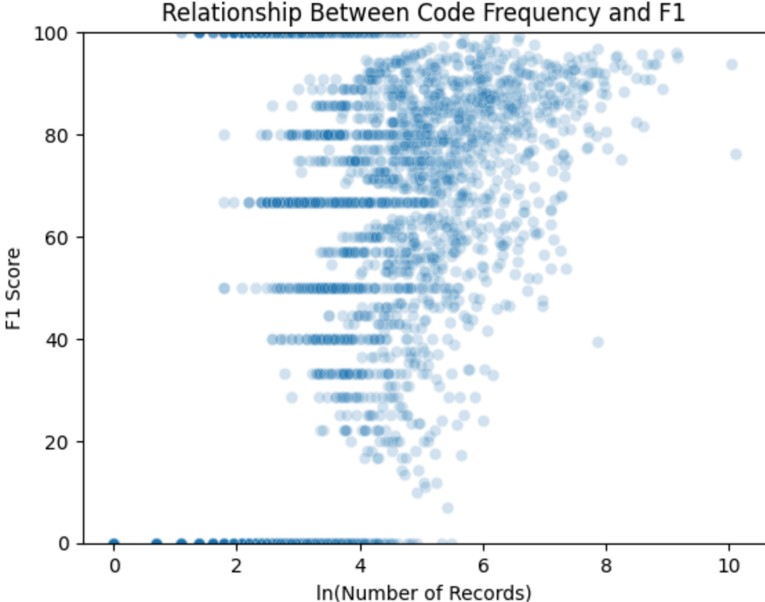

**Fig 3**. **GatorTron often achieves stronger performance on common diagnoses.** GatorTron-based F1 score for each distinct diagnosis code plotted against the total natural log frequency of each code in the full Colorado State University Veterinary Teaching Hospital labeled fine-tuning dataset. Model performance, as measured by class F1 score, was strongest for the most frequently appearing codes in the training data set. 3,906 (50.5%) codes were not represented since they did not appear in the test set. The two most frequently appearing codes were "Disease type AND/OR category unknown" (SNOMED: 3219008) and "Fit and well" (SNOMED: 102499006). The same analysis was performed for other models, and similar trends were observed.

Finally, model performance was aggregated by disease category based on the DAMNIT-V (Degenerative, Anomalous, Metabolic, Neoplastic, Nutritional, Inflammatory, Idiopathic, Toxic, Traumatic, Vascular) veterinary disease classification system (Table 5). Model performance was largely robust across disease categories, with best performance observed for the *metabolic disease* category and worst performance observed for the *nutritional disorder* category.

## Discussion

The rapidly increasing availability of pre-trained LMs [19,21,23,27,28,30,31,38,43–46] in recent years has greatly increased the ability of researchers to solve complex NLP tasks, including successful clinical coding with limited amounts of labeled fine-tuning data. The poor results (not reported here) of fine-tuning applied to several transformer models with randomly initialized weights highlighted the importance of significant pre-training for the task of clinical coding. GatorTron was pre-trained on the largest corpus of clinical data amongst all considered models (Table 1) and achieved the strongest performance after fine-tuning (Table 2). Additionally, performance on the downstream task (i.e., clinical coding) was not shown to be very sensitive to the domain of foundational LM pre-training data. General pre-training data sources have been shown to contribute to robust performance across downstream tasks [54]. The tested human clinical LMs (GatorTron, MedicalAI ClinicalBERT, and medAlpaca) and general domain LMs (BERT, RoBERTa, GPT-2, and DeBERTa V3) performed at least as well as veterinary clinical LMs (VetBERT and PetBERT).

Several other models that deserve discussion were BERT Large, DeBERTa, and ModernBERT. BERT Large was fine-tuned under a slightly different training procedure that did not feature a dropout layer. This was due to convergence issues that arose only for this model. It is speculated that BERT Large may be undertrained [44] and may not be capable of fine-tuning on the sparse data involved in this project when dropout techniques are used. DeBERTa V3 and ModernBERT both

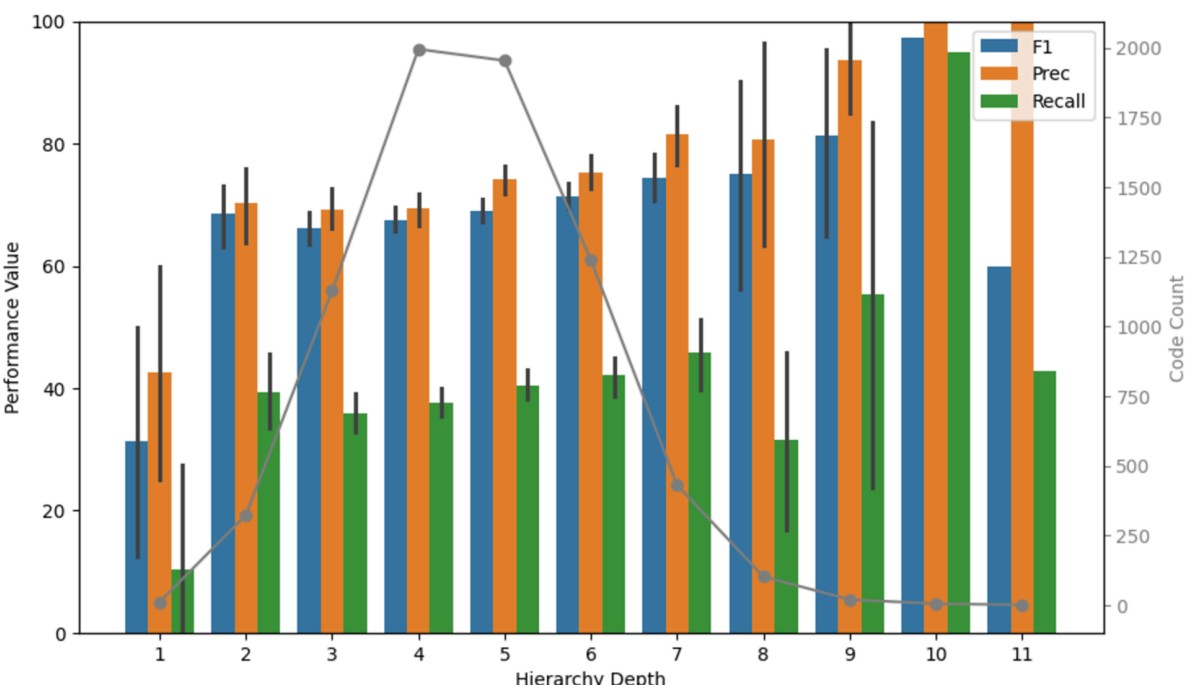

**Fig 4. GatorTron is not sensitive to code specificity.** Code depth in Systematized Medical Nomenclature for Medicine - Clinical Terms (SNOMED-CT) hierarchy (relative to the most general "Clinical Finding" at level 0) plotted against GatorTron-based class F1. A larger value for depth indicated a higher degree of specificity (e.g., a leaf code such as "Poisoning Due to Rattlesnake Venom" (SNOMED: 217659000) was at level 9). 95% confidence intervals for each level of depth are shown.

claim to be more efficient alternative model architectures but were not appreciably faster than other models (0.21 and 0.41 hrs/epoch, respectively) and did not achieve stronger performance than other comparably sized models.

Results from this study showed that the task of clinical coding was not very sensitive to model size. Relatively smaller models with respect to the total number of parameters (MedicalAI ClinicalBERT, VetBERT, PetBERT, BERT, RoBERTa, GPT-2, DeBERTa V3, ModernBERT, Clinical ModernBERT) performed comparably (5.0% worse F1 score, on average) to much larger models like GatorTron (Table 2). Further, performance was improved only relatively slightly by larger volumes of fine-tuning data, beyond a certain minimum. Accordingly, when the necessary computational resources to train relatively large LMs are unavailable, smaller ones may be adequate for this task. A sensitivity analysis of the amount of fine-tuning data used for training demonstrated that similar results could be achieved using only a small fraction (e.g., 25% – 49,978 records) of the total amount of labeled data available (Fig 5). These two findings further reduce the barriers to application of automated clinical coding by demonstrating the potential sufficiency of relatively smaller LMs and fine-tuning datasets. These are encouraging implications for situations in which limited computing resources, minimal labeled data, and reduced energy consumption are considerations [55].

While a large volume of fine-tuning data, as was used in this study, may not always be necessary for a clinical coding task, some fine-tuning is clearly helpful. "Out-of-the-box" testing performed on all selected models generally yielded poor results (Table 3). The GPT-2 models were an exception as they performed much better than other models in the "out-of-the-box" setting. Nevertheless, even the GPT-2 models were improved by more expansive fine-tuning. These findings

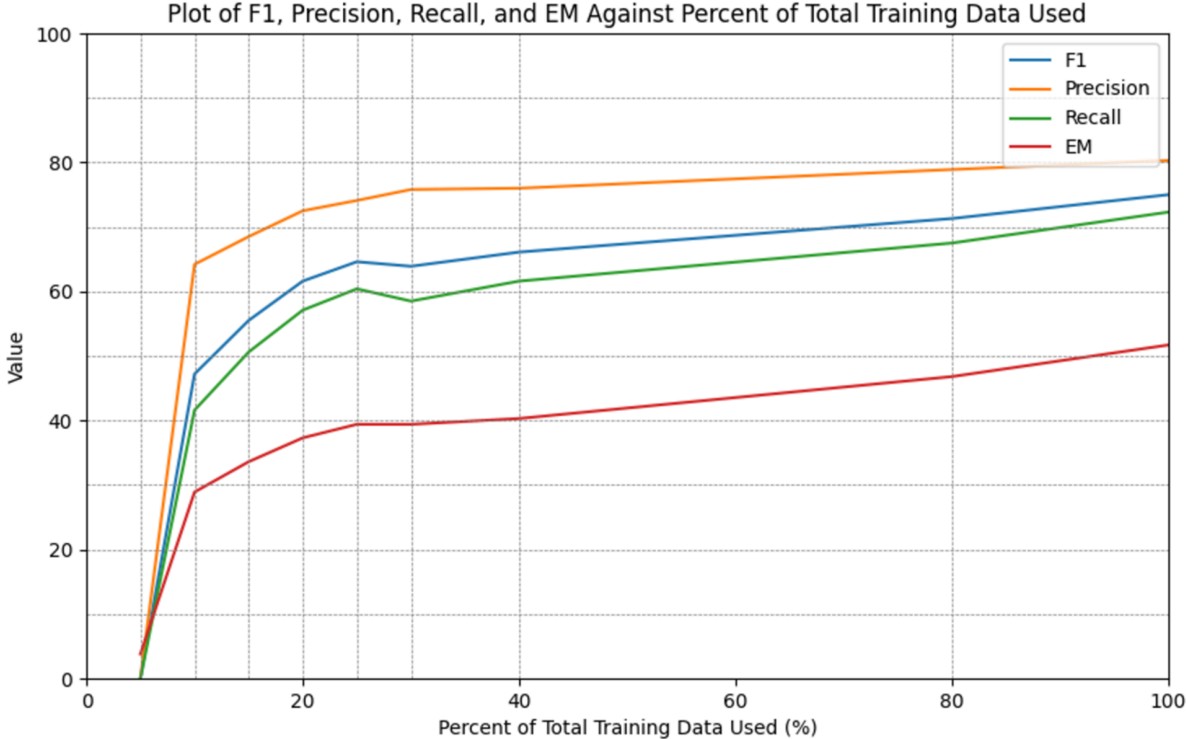

**Fig 5**. **GatorTron performance improves with additional fine-tuning data.** GatorTron-based performance metrics plotted for different percentages of total training data used in fine-tuning. Dedicated validation and test sets were used, and only the dedicated training set was successively subsetted. A stratified sampling technique, with respect to class frequencies, was used to develop subsets. 100% represents a training set with 199,914 records. The same analysis applied to other foundational models revealed similar trends.

indicated that, building on pre-trained LMs, the addition of fine-tuning with task-specific data is helpful in achieving more accurate results.

## Performance analysis

Errors made in diagnosis predictions can take many forms, and understanding the nature of these errors is important in recognizing the strengths and weaknesses of automated clinical coding and in informing future improvements. Understanding weaknesses can help prevent unsuitable application of developed tools. Several potential sources of error for the task of automated clinical coding include synonymous diagnoses recorded differently by clinicians (e.g., kidney disease and renal disease) and non-synonymous diagnoses with similar names (e.g., gingival recession and gingivitis). Additionally, determination of the acceptable level of model performance for specific clinical tasks has not yet been established.

A weakly positive relationship between frequency of code usage and predictability was observed (Fig 3). Frequently appearing codes were generally associated with stronger model performance likely because of the volume of available data that was used to learn predictive patterns. More difficult to explain is the subset of rarely appearing codes that had perfect predictive performance. This may be due to strongly consistent textual representations of a diagnosis within a small sample: if a diagnosis is described the exact same way every time it is recorded, there will be a strong predictive pattern to learn (e.g., pica, dystonia). However, many rarely appearing codes featured poor predictive performance, further indicating that class infrequency is an important factor when considering practical applications of developed models. More detailed analysis of the impact of code frequency on model prediction is warranted.

No significant relationship between code specificity and predictability was observed (Fig 4). The prevalence of codes from many depths of the SNOMED-CT hierarchy indicates that clinicians and clinical coders at the CSU VTH choose to use diagnosis concepts with varying levels of specificity. Robust model performance across much of the hierarchy testifies to the flexibility of developed automatic coding tools with respect to the specificity of clinician input. This is important as practices for recording diagnoses can often vary on a clinician-by-clinician basis. The model's generalizability across the SNOMED-CT hierarchy inspires confidence without more detailed diagnosis reporting standards being implemented. 530 codes could not be represented in this analysis as they were older, inactive codes and thus were not incorporated into the newer version of the SNOMED-CT hierarchy used in this work. This analysis was performed for multiple models, and similar trends were observed.

A positive relationship between volume of data used for fine-tuning and model performance was observed (Fig 5). This multi-classification task is especially suited to large volumes of fine-tuning due to the large number of classes (7,739) involved and the relative rarity of a large number of the classes. However, model performance was found to be relatively more dependent when smaller amounts of fine-tuning data were used. Potentially useful model performance can be achieved with even a small fraction (e.g., 25%) of the CSU VTH fine-tuning dataset, with diminishing marginal returns for additional data. Certainly, the use of more labeled data for fine-tuning yields superior results but having a labeled fine-tuning dataset smaller than the one from CSU VTH may still be useful. It should be noted that the sensitivity of this finding to factors like model size and domain of pre-training data has not yet been determined. This is a significant finding for sites without an existing large, well-curated, fine-tuning dataset where labeled data may have to be assembled from scratch.

Largely robust model performance was observed across high-level disease categories (Table 5). Understanding the strengths and weaknesses of developed models with respect to disease domains is crucial for responsible application. Only the *nutritional disorder* category demonstrated appreciably lower performance and was also the category with the fewest observations. While initial results showed encouraging robustness, many diagnosis codes used in this project have yet to be properly classified. The structure of the SNOMED-CT hierarchy does not allow for all codes of interest to be easily mapped to one of the selected high-level disease categories. For example, a manual mapping approach will be required to categorize diagnoses currently in the *other* category.

## Applications

A fast and reliable automated clinical coding tool has significant applications in the curation of high-quality data for clinical and research purposes. One of the primary motivations of this project was the loading of standardized diagnosis codes

**Table 5**. GatorTron performance on common disease categories.

| Category | N | F1 | Precision | Recall |
|---|---|---|---|---|
| Other | 174,789 | 77.0 | 82.4 | 73.9 |
| Neoplasm and/or hamartoma | 64,170 | 76.4 | 83.5 | 72.0 |
| Inflammatory disorder | 49,261 | 82.9 | 88.0 | 79.7 |
| Traumatic or non-traumatic injury | 27,710 | 70.2 | 79.3 | 65.2 |
| Degenerative disorder | 23,132 | 81.4 | 88.6 | 76.2 |
| Disorder of cardiovascular system | 21,946 | 78.9 | 86.7 | 74.4 |
| Metabolic disease | 11,942 | 90.8 | 95.9 | 86.9 |
| Congenital disease | 7,802 | 76.6 | 84.1 | 72.1 |
| Poisoning | 1,044 | 79.7 | 84.9 | 75.6 |
| Nutritional disorder | 669 | 54.8 | 79.2 | 43.8 |

GatorTron-based performance metrics aggregated by exclusive disease categories defined by the DAMNIT-V classification system for disease categories.

inferred from free-text inputs into an OMOP CDM instance. The developed model demonstrated the potential to perform large-scale automated coding.

However, naive reliance on AI approaches to assemble research-quality data is problematic. Considerable confidence-based selectivity, manual validation, and data quality checks are required to preserve the integrity of an OMOP CDM instance, particularly when AI is introduced into the coding process. Therefore, other applications that more intentionally preserve human-in-the-loop validation must be considered. The developed tool could serve as a supplemental assistant to the existing medical records staff at veterinary hospitals. Rather than having to search for appropriate codes from scratch, coders could have a list of suggested codes supplied to them at the onset and could accept, reject, or augment the suggested codes as needed. This may improve the agility of manual coding and would still assist in the curation of data for storage in an OMOP CDM instance.

It is further considered that clinicians themselves could serve as human-in-the-loop coders as part of the process of finalizing records, given their expertise in the field. This has not yet been implemented because, based on discussions with professionals about this idea at CSU, clinicians have limited time to manually search through unfamiliar medical code lists to find appropriate concepts. However, if the developed automated coding tool were to quickly and reliably suggest diagnosis codes at the point of textual entry, clinicians could more easily accept or reject suggestions and proceed with their work. This would serve as a front-line coding practice that could capture a significant portion of diagnoses at the point of entry and the remaining uncoded diagnoses could be passed to expert coding staff for further review.

Ongoing research at the CSU VTH aims to demonstrate speed and reliability increases when using AI assistance to aid professional medical coders and clinicians in clinical coding tasks. Regardless of the potential approaches that are used to implement automated coding into the EHR standardization process, great care must be taken to ensure that high data quality standards are achieved and maintained. Rigorous and continual quality assurance practices will be needed to ensure the lasting quality of model predictions.

## Future work

A major challenge to the growth of widespread use of AI in veterinary medicine is trust, both by clinical professionals and the public [56]. Evaluative tasks that help engender trust and confidence in model predictions will be included in future work in support of eventual applicability in a clinical setting. An initial approach to model explainability used the integrated gradients method [57] to attribute predictive importance to input tokens. Similar methods have been implemented in other studies [58]. Future work will continue to expand on these efforts. Other tasks include (1) performing in-depth error analyses by examining the source text in collaboration with clinical experts (e.g., additional performance analyses), (2) demonstrating the generalizability of the model to applications beyond CSU (e.g., testing models against data from private practices or other academic veterinary institutions), (3) introducing new sources of textual data to model inputs (e.g., patient history, medication lists) and (4) selecting models with larger context windows [38,59,60], and exploring more complex and potentially stronger [61] modeling processes (e.g., ensemble approaches).

The field of AI has been developing at an extraordinary pace. Since the inception of this project, several new foundational clinical language models have been released that could challenge the primacy of GatorTron. Yale University's Me-LLaMA [21] and the Swiss Federal Institute of Technology Lausanne's MEDITRON [27] have both been published since 2023. Due to computational limitations, neither of these models were directly considered for this project. However, future research in this field should leverage all possible foundational LMs to remain cutting edge and determine the best solution for automating clinical diagnosis coding.

## Acknowledgments

The authors thank the following entities and individuals: the Colorado State University Veterinary Teaching Hospital medical records team for their invaluable contributions to data curation, the Colorado State University Data Science Research

Institute for providing access to critical high performance computing resources, and Bill Carpenter for providing support and oversight for the use of aforementioned computing resources.

## Author contributions

**Conceptualization:** Mayla R. Boguslav, Adam Kiehl, David Kott, George Joseph Strecker, Michael Kirby.

**Data curation:** Terri Ward.

**Formal analysis:** Mayla R. Boguslav, Adam Kiehl.

**Funding acquisition:** George Joseph Strecker, Michael Kirby.

**Investigation:** Mayla R. Boguslav, Adam Kiehl, David Kott.

**Methodology:** Mayla R. Boguslav, Adam Kiehl, David Kott, George Joseph Strecker, Michael Kirby.

**Project administration:** George Joseph Strecker, Michael Kirby.

**Resources:** Michael Kirby.

**Software:** Mayla R. Boguslav, Adam Kiehl, David Kott.

**Supervision:** George Joseph Strecker, Michael Kirby.

**Validation:** Mayla R. Boguslav, Adam Kiehl.

**Visualization:** Adam Kiehl.

**Writing – original draft:** Mayla R. Boguslav, Adam Kiehl, Tracy L. Webb, Nadia Saklou.

**Writing – review & editing:** Mayla R. Boguslav, Adam Kiehl, David Kott, George Joseph Strecker, Tracy L. Webb, Nadia Saklou, Michael Kirby.

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
