## [Decision Letter · Decision Letter 0]

17 Apr 2025

PDIG-D-24-00525Fine-tuning foundational models to code diagnoses from veterinary health recordsPLOS Digital Health
Dear Dr. Adam Kiehl,
Thank you for submitting your manuscript to PLOS Digital Health. After careful consideration, we feel that it has merit but does not fully meet PLOS Digital Health's publication criteria as it currently stands. Therefore, we invite you to submit a revised version of the manuscript that addresses the points raised during the review process. Please submit your revised manuscript within 60 days Jun 16 2025 11:59PM. If you will need more time than this to complete your revisions, please reply to this message or contact the journal office at digitalhealth@plos.org. Please include the following items when submitting your revised manuscript:* A rebuttal letter that responds to each point raised by the editor and reviewer(s). You should upload this letter as a separate file labeled 'Response to Reviewers'. This file does not need to include responses to any formatting updates and technical items listed in the 'Journal Requirements' section below.* A marked-up copy of your manuscript that highlights changes made to the original version. You should upload this as a separate file labeled 'Revised Manuscript with Track Changes'.* An unmarked version of your revised paper without tracked changes. You should upload this as a separate file labeled 'Manuscript'. If you would like to make changes to your financial disclosure, competing interests statement, or data availability statement, please make these updates within the submission form at the time of resubmission. Guidelines for resubmitting your figure files are available below the reviewer comments at the end of this letter. We look forward to receiving your revised manuscript. Kind regards, Xiaoli Liu, PhDAcademic EditorPLOS Digital Health Xiaoli LiuAcademic EditorPLOS Digital Health Leo Anthony CeliEditor-in-ChiefPLOS Digital Healthorcid.org/0000-0001-6712-6626**Journal Requirements:**

1. We ask that a manuscript source file is provided at Revision. Please upload your manuscript file as a .doc, .docx, .rtf or .tex. **Additional Editor Comments (if provided):** Dear Dr. Adam Kiehl,

Thank you for your earlier suggestions regarding potential reviewers. While we have made concerted efforts to identify qualified experts in this specialized field, many potential candidates were unavailable due to scheduling conflicts. We have now received substantive feedback from two reviewers whose perspectives align well with the technical scope of your work. I would strongly encourage your team to systematically address each point raised in these evaluations through both textual revisions and point-by-point responses in your rebuttal letter.

Please note that our editorial team continues to explore additional reviewer options to ensure comprehensive evaluation. Should we secure further expert insights during this ongoing process, we will promptly communicate any supplementary feedback to facilitate your manuscript's refinement. Thank you for your understanding and time!

Best,

Dr. Liu**Reviewers' Comments:** Reviewer's Responses to Questions

**Comments to the Author**

1. Does this manuscript meet PLOS Digital Health’s publication criteria? Is the manuscript technically sound, and do the data support the conclusions? The manuscript must describe methodologically and ethically rigorous research with conclusions that are appropriately drawn based on the data presented.

Reviewer #1: Partly

Reviewer #2: Yes

2. Has the statistical analysis been performed appropriately and rigorously?

Reviewer #1: Yes

Reviewer #2: Yes

3. Have the authors made all data underlying the findings in their manuscript fully available (please refer to the Data Availability Statement at the start of the manuscript PDF file)?

Reviewer #1: No

Reviewer #2: No

4. Is the manuscript presented in an intelligible fashion and written in standard English?

Reviewer #1: Yes

Reviewer #2: Yes

5. Review Comments to the Author

Reviewer #1: Summary of the Paper

The authors present a paper that investigates the use of transformer-based language models to automate clinical coding of diagnoses from veterinary electronic health records (EHRs). They draw on 246,473 manually coded patient visits from a single university’s veterinary teaching hospital, aiming to categorize free-text clinical summaries into one or more of 7,739 SNOMED-CT diagnosis concepts. The study compares a broad array of models, ranging from smaller pre-trained transformers (e.g., BERT, GPT-2) to more extensive clinical and veterinary-specific architectures (GatorTron, medAlpaca, VetBERT, PetBERT). Results indicate that fine-tuned versions of GatorTron outperform other models, achieving an F1 score above 70% and exact match of roughly 50%. The authors also analyze how factors such as code frequency, SNOMED hierarchy depth, and training data volume affect the model’s coding accuracy, thereby exploring real-world considerations like the interpretability and efficiency of automated coding processes. They emphasize integration of standardized vocabularies (SNOMED-CT) and data models (OMOP CDM) to further interoperability and One Health–driven analytics in veterinary medicine.

Strengths

1. Comprehensive Model Comparison

The paper systematically evaluates a range of transformer-based models, from smaller general-purpose variants (BERT, GPT-2) to larger, domain-oriented ones (GatorTron, medAlpaca), and veterinary-specific architectures (VetBERT, PetBERT). This holistic approach highlights which strategies and pre-training corpora might best suit the veterinary coding task.

2. Large Dataset

The dataset of 246K manually coded veterinary visits is a substantial resource, enabling robust fine-tuning. Few studies in the veterinary domain leverage such an extensive labeled corpus, thereby enhancing the credibility and statistical strength of the results.

3. Use of SNOMED-CT

By employing the SNOMED-CT vocabulary for diagnoses, the authors align their methodology with established clinical and research standards. This choice supports interoperability with both human and veterinary records—a vital consideration for One Health frameworks.

4. Thorough Evaluation Metrics

The inclusion of precision, recall, F1 score, and exact match offers a comprehensive look at multi-label classification performance. Exact match, in particular, is a stringent measure relevant for assigning multiple correct codes to a single clinical note.

5. Real-World Applicability

The paper situates its approach in a practical setting, discussing how automated coding can expedite and standardize workflows in veterinary informatics. The results, especially those for fine-tuned GatorTron, suggest that such methods could be deployed in production environments for clinical or research support.

Areas for Improvement

1. Terminology and Model Scale

Although the authors refer to all tested architectures as “large language models,” many (e.g., BERT base, GPT-2) fall into the smaller-to-mid-scale range by current standards. It would be more accurate to label them generally as pre-trained transformer models or small language models and reserve “LLM” for models of substantially larger parameter counts.

2. Choice of Input Sections

The paper focuses on the “Diagnosis” and “Assessment” EHR fields without deeply justifying why other sections (e.g., lab findings, treatment notes) are excluded. Additional sections might contain critical diagnostic indicators. A brief rationale for this decision, or experiments demonstrating that these two sections suffice for accurate classification, would strengthen the manuscript’s methodological transparency.

3. Tokenization and Truncation Strategy

Some portion of the dataset (up to 13% of notes) was truncated to fit within the models’ input limits, but the authors do not analyze how much this truncation might harm performance. Past research (e.g., Beltagy et al. 2020 on Longformer) demonstrates that transformer models with extended context windows often outperform those that must heavily truncate clinical notes. Examining alternative or more advanced approaches (e.g., chunking, hierarchical attention) would be beneficial. In the case of this paper, only 13% of notes were truncated, but in medicine, it is very common for notes to be longer than some of the limited context size.

4. Fine-Tuning Details

The paper provides limited information on whether all layers were fine-tuned or only top layers. Techniques like layer-wise learning rate decay or partial freezing can impact performance and efficiency. More explicit detail on learning rates, epochs, and potential experiments comparing different fine-tuning strategies would aid reproducibility. This is particularly true for your “out-of-the-box” comparison, as you have to attach a classifier head, which means you are technically fine-tuning, just freezing the transformer layers.

5. Explainability of Model Outputs

While the paper thoroughly reports predictive metrics, it lacks any discussion of interpretability methods. Explainability is crucial in medical or veterinary contexts to build trust and identify potential errors in model reasoning. Approaches such as attention visualization, Integrated Gradients, LIME, or SHAP (as in Mullenbach et al. 2018) could help users understand how the model arrived at a specific set of codes.

6. Out of the box

While the paper justifies the “out-of-the-box” comparison, freezing the weights and only implementing a classifier head, there has been significant research into how this creates unstable gradients and unpredictable (and challenging to reproduce) experiments. It is not surprising, and a very well-known scenario, that encoder models are used for different purposes, and each mode has different recommended pre-training and fine-tuning methods. Some of the failures of convergence are most likely related to these behaviors.

7. Category-Level Error Analysis

The paper does not present a detailed breakdown of which SNOMED-CT categories are more prone to errors. Analyses of precision/recall by diagnostic group (e.g., musculoskeletal, gastrointestinal, etc.) would reveal if the model struggles disproportionately with certain code sets. Understanding these weaknesses is often critical for targeted improvements and safer clinical adoption.

Recommendations for Improvement

1. Clarify Model Terminology

Differentiate between large-scale models (with billions of parameters) and smaller or mid-sized transformer-based architectures. Reserve the term “LLM” for the former, and use “pre-trained transformer” or “foundation model” more generally. Most of the work the paper could use would be “medium” at best. There is also a little discussion on MoE type models (albeit due to size) and “agentic” style models that should be discussed early on so users understand that you generally focus only on SLM/pre-trained models of the encoder only variety.

2. Define the Fine-tuning process

The fine-tuning methodology is unclear, with the only statement being as many as possible. There is a lot of work out there about how to fine-tune the non-vet models (i.e. VetBert and PetBert are not widely used), and the decision of how to or how not to fine-tune the model is critical for the experiment. Justification also needs to be given as to why, for the non-veterinary models, pretraining was not done. Some important questions needing answers:

a. How many layers were the final models fine-tuned on?

b. How many unknown/sub-word tokens per a model?

c. Progressive unfreezing was not used because?

d. Total number of epochs

e. Gradient accumulation and gradient clipping parameters

f. And a few others.

3. Address Truncation Effects

Conduct a brief study of how truncating the text impacts coding accuracy. If truncation substantially degrades performance, adopting transformer architectures that can handle longer context windows—such as Longformer (Beltagy et al. 2020)—or chunking strategies might be beneficial.

4. Incorporate Explainability Tools

The authors have already started discussing the relationship of F1 score to prediction terms. Still, word cloud analysis, syllable/complexity analysis, and other standard explainability tools (LIME/SHAP) should also be considered to understand why the algorithms output their predictions. Particularly if there are patterns of input that seem to result in different outputs.

5. Error Analysis by Diagnosis Category

Present an analysis that groups SNOMED-CT codes into broader categories and reports classification metrics for each. It is not necessary to do an “in-depth” analysis on this part, but if there are clusters, particularly around veterinary only or human diseases that behave differently in animals, this would be important. There is a lot of work around noun-confusion in transformers and knowing this is a critical component to understanding performance.

6. Consider not using a pooler

Using the pooler layer, may result in poor performance independent of mode. Pooler layers have weights and biases that are directly related to how the "Vanilla" model was pre-trained and it's downstream tasks. I would recommend not using it for 2 reasons, 1) not all models have it so additional future comparison is challenging and 2) the biases are a known issue and impact downstream task adoption. Consider using a FC with a normalization layer to the last hidden state of the transformer model for a more fair comparison.

7. Use ModerBert or DeBerta based architectures (among other options like distilled biomistral)

There has been a lot of work in the past year or so on architectures that change how attention is utilized and have significantly improved performance beyond vanilla BERT. He, et al 2023 for DeBERTaV3 and Warner, et al 2024 for ModernBERT show many of the flaws that current BERT architectures has and particularly the problem you are experiencing in the analysis are discussed in both papers. I strongly encourage the authors to reconsider their model choices as while veterinary medicine is “new,” there has been much work in this area already, and we can leverage that to skip the issues. I would reduce the number of models tested and improve the depth of analysis as the broad stroke process for known poor-performance (and vanilla out of the box) reduces the potential paper's impact.

Additional References to Review and Incorporate

• Explainability in Clinical NLP Mullenbach et al. 2018

• Long Document Transformers for Medical Text Beltagy et al. 2020

• Smarter, Better, Faster, Longer: A Modern Bidirectional Encoder for Fast, Memory Efficient, and Long Context Finetuning and Inference Warner et al. 2024.

• DeBERTaV3: Improving DeBERTa using ELECTRA-Style Pre-Training with Gradient-Disentangled Embedding Sharing He et al. 2023

• Transformer models in biomedicine Madan et al. 2024

• Task-Specific Transformer-Based Language Models in Health Care: Scoping Review Cho, et al. 2024

• Transformer Models in Healthcare: A Survey and Thematic Analysis of Potentials, Shortcomings and Risks Denecke et al 2024

• A comparative study of pretrained language models for long clinical text Li et al. 2022

• TransformEHR: transformer-based encoder-decoder generative model to enhance prediction of disease outcomes using electronic health records Yang et al. 2023

• A Pretrainer’s Guide to Training Data: Measuring the Effects of Data Age, Domain Coverage, Quality, & Toxicity Longpre et al. 2024

Minor corrections:

1. Some of the sentences are particularly long with the usage of i.e. Consider removing the i.e. on repeated usage or in many cases since the paper is trying to generalize, use e.g. as the preferred term.

2. Figure 2 text is very long, describing a “result” of success on that note. While interesting, the point of Figure 2 is to show the “data.” It may be beneficial to show examples of cases that all algorithms could get when they were vanilla vs fine-tuned, but this would be nice to have.

3. On Page 18, The graph has In() number of records, is there supposed to be a unit, like x1k records? If not, remove it

4. On page 19, the Figure 4 definition is super long. Some of that text should be in the discussion of the figure and not in the figure text. Consider shortening the Figure 4 description and moving the valuable discussion to the discussion text directly

Reviewer #2: It's understandable that the data used to train the model cannot be made available, but the model itself or any ability to evaluate the model should be made available, as was the case for both VetBERT and PetBERT. Otherwise, the study is only pointing to a model that is theoretically better than others without any way to evaluate. At a minimum there should be a section explaining why the model is not being made available.

6. PLOS authors have the option to publish the peer review history of their article (what does this mean?). If published, this will include your full peer review and any attached files.

**Do you want your identity to be public for this peer review?** For information about this choice, including consent withdrawal, please see our Privacy Policy.

Reviewer #1: **Yes:** Jonathan L Lustgarten

Reviewer #2: No

---

## [Decision Letter · Decision Letter 1]

4 Dec 2025

Fine-tuning foundational models to code diagnoses from veterinary health records

PDIG-D-24-00525R1

Dear Kiehl,

We are pleased to inform you that your manuscript 'Fine-tuning foundational models to code diagnoses from veterinary health records' has been provisionally accepted for publication in PLOS Digital Health.

Best regards,

Mathew V. Kiang, PhD

Section Editor

PLOS Digital Health

**Additional Editor Comments (if provided):**

Dear Dr. Adam Kiehl,

We are thrilled to have received feedback from all the reviewers. Excellent work—we look forward to sharing it with the readers promptly!

Cheers!

Dr. Liu

**Reviewer Comments (if any, and for reference):**

Reviewer's Responses to Questions

**Comments to the Author**

1. If the authors have adequately addressed your comments raised in a previous round of review and you feel that this manuscript is now acceptable for publication, you may indicate that here to bypass the “Comments to the Author” section, enter your conflict of interest statement in the “Confidential to Editor” section, and submit your "Accept" recommendation.

Reviewer #3: All comments have been addressed

2. Does this manuscript meet PLOS Digital Health’s publication criteria? Is the manuscript technically sound, and do the data support the conclusions? The manuscript must describe methodologically and ethically rigorous research with conclusions that are appropriately drawn based on the data presented.

Reviewer #3: Yes

3. Has the statistical analysis been performed appropriately and rigorously?

Reviewer #3: Yes

4. Have the authors made all data underlying the findings in their manuscript fully available (please refer to the Data Availability Statement at the start of the manuscript PDF file)?

Reviewer #3: Yes

5. Is the manuscript presented in an intelligible fashion and written in standard English?

Reviewer #3: Yes

6. Review Comments to the Author

Reviewer #3: I think the authors have adequately addressed the comments from the previous reviewers.

On the other hand, it could be beneficial to conduct a mini study (It is completely optional to be included in this manuscript) on how confident the clinicians are to use assistive AI tools and accept the answers from the LMs in their practices, if the LMs are to be pushed into testing phase.

7. PLOS authors have the option to publish the peer review history of their article (what does this mean?). If published, this will include your full peer review and any attached files.

**Do you want your identity to be public for this peer review?** For information about this choice, including consent withdrawal, please see our Privacy Policy.

Reviewer #3: No
